# Technical Note: Altitude scaling of $^{36}$Cl production from Fe

Angus K. Moore[1,2], Darryl E. Granger[1]

[1]Department of Earth, Atmospheric, and Planetary Sciences, Purdue University
 550 Stadium Mall Drive
 West Lafayette, IN 47907, USA
[2]Now at Georges Lemaître Centre for Earth and Climate Research, Earth and Life Institute, UCLouvain
 Place Louis Pasteur 3
 1348 Louvain-la-Neuve, Belgium

*Correspondence to*: Angus K. Moore (angus.moore@uclouvain.be)

**Abstract.** Cosmogenic nuclide production rates depend on the excitation functions of the underlying nuclear reactions and the intensity and energy spectrum of the cosmic ray flux. The cosmic ray energy spectrum shifts towards lower average energies with decreasing altitude (increasing atmospheric depth), so production rate scaling will differ for production reactions that have different energy sensitivities. Here, we assess the possibility of unique scaling of $^{36}$Cl production from Fe by modeling changes in the $^{36}$Cl$_{Fe}$/$^{36}$Cl$_{K}$ and $^{36}$Cl$_{Fe}$/$^{10}$Be$_{qtz}$ production ratios with altitude. We evaluate model predictions against measured $^{36}$Cl concentrations in magnetite and K-feldspar and $^{10}$Be concentrations in quartz from granitic rocks exposed across an elevation transect (ca. 1700-4300 m asl) in western North America. The data are broadly consistent with model predictions. The null hypothesis, that $^{36}$Cl$_{Fe}$/$^{10}$Be$_{qtz}$ and $^{36}$Cl$_{Fe}$/$^{36}$Cl$_{K}$ production ratios are invariant with altitude, can be rejected at the 90% confidence level. Thus, reaction-specific scaling factors will likely yield more accurate results than non-reaction-specific scaling factors when scaling $^{36}$Cl production in Fe-rich rocks and minerals.

## 1. Introduction

Cosmogenic nuclides are produced in rock exposed at the surface of the Earth through nuclear reactions between cosmic rays and target nuclei in the rock. These *in situ*-produced cosmogenic nuclides give rise to a family of geochronologic systems that can be used to determine surface exposure ages, erosion rates, and burial dates of late-Cenozoic sediments (von Blanckenburg and Willenbring, 2014; Granger et al., 2013). General application of cosmogenic nuclide geochronology across the surface of the Earth requires modeling how cosmogenic nuclide production rates scale in space and time due to geomagnetic modulation of the primary cosmic ray flux and attenuation of cosmic radiation in the atmosphere.

The most important cosmogenic nuclide production reactions are high-energy nucleon (neutron and proton) spallation. The production rate of a cosmogenic nuclide by spallation at any location on Earth's surface depends on the excitation functions of the underlying nuclear reactions (i.e., the cross section, a measure of reaction probability, as a function of incident particle energy) and the intensity and energy distribution of the cosmic radiation at that location. Monte Carlo simulations of the atmospheric nucleonic cascade show that nucleon energy spectra shift towards lower average energies with decreasing elevation (Argento et al., 2015a, 2015b, 2013). This implies that the production ratio between two reactions with different energy sensitivities change with altitude.

Several studies have investigated changing production ratios with altitude. For example, the $^{3}$He/$^{10}$Be ratio has been observed to increase with altitude in the Himalaya (Amidon et al., 2008; Gayer et al., 2004), although a similar signal is ambiguously absent in datasets from the Andes and Mt. Kilimanjaro (Blard et al., 2013; Schimmelpfennig et al., 2011). Likewise, Corbett et al. (2017) measured an $^{26}$Al/$^{10}$Be ratio in quartz at sea level and high latitude (SLHL) that is approximately 7% higher than the conventional ratio of ca. 6.8 originally calibrated at high elevation and mid-latitude (Nishiizumi et al., 1989). Global analysis of $^{26}$Al/$^{10}$Be ratio data in quartz also supports spatial variation in this production ratio (Halsted et al., 2021). However,

because the reactions producing $^{10}$Be and $^{26}$Al in quartz are sensitive to the lower end of the high-energy spectrum, the signal is subtle and may be easily overprinted by, for example, inherited muon-produced nuclides.

Changes in production ratio with altitude should be most clearly resolvable between two reactions that are widely separated in energy sensitivity. Cosmic ray energy spectra below approximately 500 MeV, the maximum energy associated with secondary recoil nucleons, change little with elevation because the slowing down of secondary nucleons from this energy to rest occurs essentially locally in the atmosphere (Lal and Peters, 1967). This means that all reactions sensitive primarily to energies below 500 MeV should exhibit only modest differences in scaling with altitude, regardless of the shape of the underlying excitation functions, whereas reactions that peak above 500 MeV should be more rapidly attenuated in the atmosphere.

Of the commonly measured *in situ*-produced cosmogenic nuclides, $^{36}$Cl offers the widest range of practical production pathways, including spallation from four major rock-forming elements (K, Ca, Ti, and Fe). These reactions encompass a wide range of energy sensitivities and can be isolated by analyzing mineral separates with tightly defined target chemistries. Of the four reactions, spallation from Fe has the highest threshold and peak (Schiekel et al., 1996; Reedy, 2013) (Figure 1), and thus is most likely to exhibit a clearly detectable departure from the other $^{36}$Cl production pathways. Previous work studied the $^{36}$Cl production rate from Fe in magnetite relative to $^{10}$Be in quartz (Moore and Granger, 2019a), although it did not conclusively confirm or refute reaction-specific scaling. In that study, the difference in time-integrated geomagnetic cutoff rigidity between widely separated calibration sites may have offset the effect of a ca. 1800 m elevation difference.

Here, we directly evaluate reaction-specific scaling of $^{36}$Cl production from Fe by examining changes in $^{36}$Cl$_{Fe}$/$^{36}$Cl$_K$ and $^{36}$Cl$_{Fe}$/$^{10}$Be$_{qtz}$ production ratios with altitude. First, ratios are modeled as a function of altitude using cosmic ray energy spectra in conjuncture with available excitation functions. Model predictions are then tested against data from Moore and Granger (2019a) and new measurements of $^{36}$Cl in magnetite and K-feldspar and $^{10}$Be in quartz from granitic boulders exposed at elevations of ca. 1700 and 4300 m asl, but similar time-integrated cutoff rigidities, in western North America. The results have implications for accurately estimating $^{36}$Cl production rates in Fe-rich rocks (e.g., peridotite or basalt) and mineral separates (e.g., magnetite), which are increasingly used to determine erosion rates and exposure ages in quartz-poor mafic and ultramafic landscapes (e.g., Leontaritis et al., 2022; Moore et al., 2024; Moore and Granger, 2019a).

## 2. Methods

### 2.1 Production model

To predict the scaling behavior of $^{36}$Cl production from Fe, we model production rates across an altitude transect. Production rates are calculated by integrating the product of the reaction cross section and particle flux across all cosmic ray energies (Lal and Peters, 1967):

$$P_i = \sum_j N_j \sum_k \int_0^\infty \frac{d\Phi_k(E_k)}{dE_k} \sigma_{ijk}(E_k) dE_k \qquad \text{Eqn. 1}$$

Where $P_i$ is the production rate of nuclide i (in atoms g$^{-1}$ yr$^{-1}$), $N_j$ is the number of target nuclei j per gram of target material, $\sigma_{ijk}(E_k)$ is the cross section for production of nuclide i on target atom j, by particle k at energy level $E_k$, and $\Phi_k(E_k)$ is the omnidirectional flux of particles of type k of energy E that are incident at the location of interest (i.e., $d\Phi_k(E_k)/dE_k$ is the differential energy spectrum). Cross sections are typically derived from irradiation experiments (e.g., Schiekel et al., 1996), whereas particle fluxes are usually estimated from Monte Carlo modeling of the cosmic ray cascade (e.g., Masarik and Beer, 2009).

Direct calculation of cosmogenic nuclide production rates using Eqn. 1 has historically been impeded, in part, by the high computational costs of Monte Carlo modeling of the evolution of the cosmic ray cascade through the atmosphere. The PARMA model (Sato et al., 2008), which drives the widely applied Lifton-Sato-Dunai scaling model for *in situ* production rates (Lifton et al., 2014), provides a computationally efficient way of estimating cosmic ray intensities and energy spectra using analytical

functions fit to the output of Monte Carlo simulations. Precision in the nucleon spectra from PARMA is similar to that derived directly from Monte Carlo models (Sato et al., 2008). Statistical uncertainties in predicted nucleons fluxes tend to increase towards sea level, where they are estimated at 20%; however, the model fits available particle intensity observations with significantly better fidelity (Lifton et al. 2014), especially for the higher-energy nucleonic component of the cascade. We use the PARMA model as implemented in Lifton et al. (2014) to model $\Phi_k(E_k)$. Figure 1 illustrates the modeled increase in intensity of the high energy portion of the cosmic ray nucleon spectra with increasing altitude.

Excitation functions are taken from the compilation of Reedy (2013) and consist of evaluated cross sections compiled chiefly from the CSISRS cross section database maintained by Brookhaven National Laboratory. The excitation function for $^{36}Cl$ production from Fe by protons was measured by Schiekel et al., (1996). The estimate of the excitation function for the corresponding neutron reaction is broadly similar, with a slightly lower threshold (Figure 1). This is consistent with the tendency of neutron and proton cross sections to be comparable at high energies. Furthermore, $^{36}Cl$ is intermediate in mass between two stable isotopes, $^{35}Cl$ and $^{37}Cl$. Neutron and proton reactions that make a nuclide between two stable nuclides often have similar excitation functions (Reedy, 2013).

**2.2 Calibration samples**

We examine production ratios across three sites that span a ca. 2600 m elevation transect at mid-latitude in western North America (Figure 2, Table 1). Three samples were collected at the low elevation site, a moraine located at the mouth of Pine Creek Canyon in Owens Valley, California at ca. 1700 m asl. This moraine corresponds to the Tioga 2 stage of glaciation in the Sierra Nevada between ca. 25-20 ka (Phillips et al., 2009). The mid-elevation samples (collected from two sites at approximately 3300 m asl) were previously described in Moore and Granger (2019a). These samples were collected from the surfaces of erratic boulders located behind late-glacial Recess Peak (13.3 ka) and Tioga 4 (15.8 ka) moraines in the Sierra Nevada of California (Phillips, 2016). A legacy sample, collected in 1997 by David Elmore from ca. 4300 m asl on Mt. Evans in Colorado, forms the uppermost member of the transect. This sample was situated near the Mt. Evans summit parking lot in an unglaciated alpine blockfield that we model as eroding in steady state. The sample consists of a whole boulder that was crushed in 1997. No record survives of the initial geometry of the boulder. Therefore, we estimate the sample thickness from the total sample mass, approximating the boulder geometry as a cube (Table 1). The uncertainty introduced by this approach is unlikely to significantly affect the results of the analysis because production ratios are sensitive to sample thickness only insofar as different production mechanisms have different attenuation lengths or radioactive decay is significant. For Owens Valley and Mt. Evans, magnetite, K-feldspar, and quartz separates were analyzed, whereas only magnetite and quartz were examined at the mid-elevation Sierra Nevada sites.

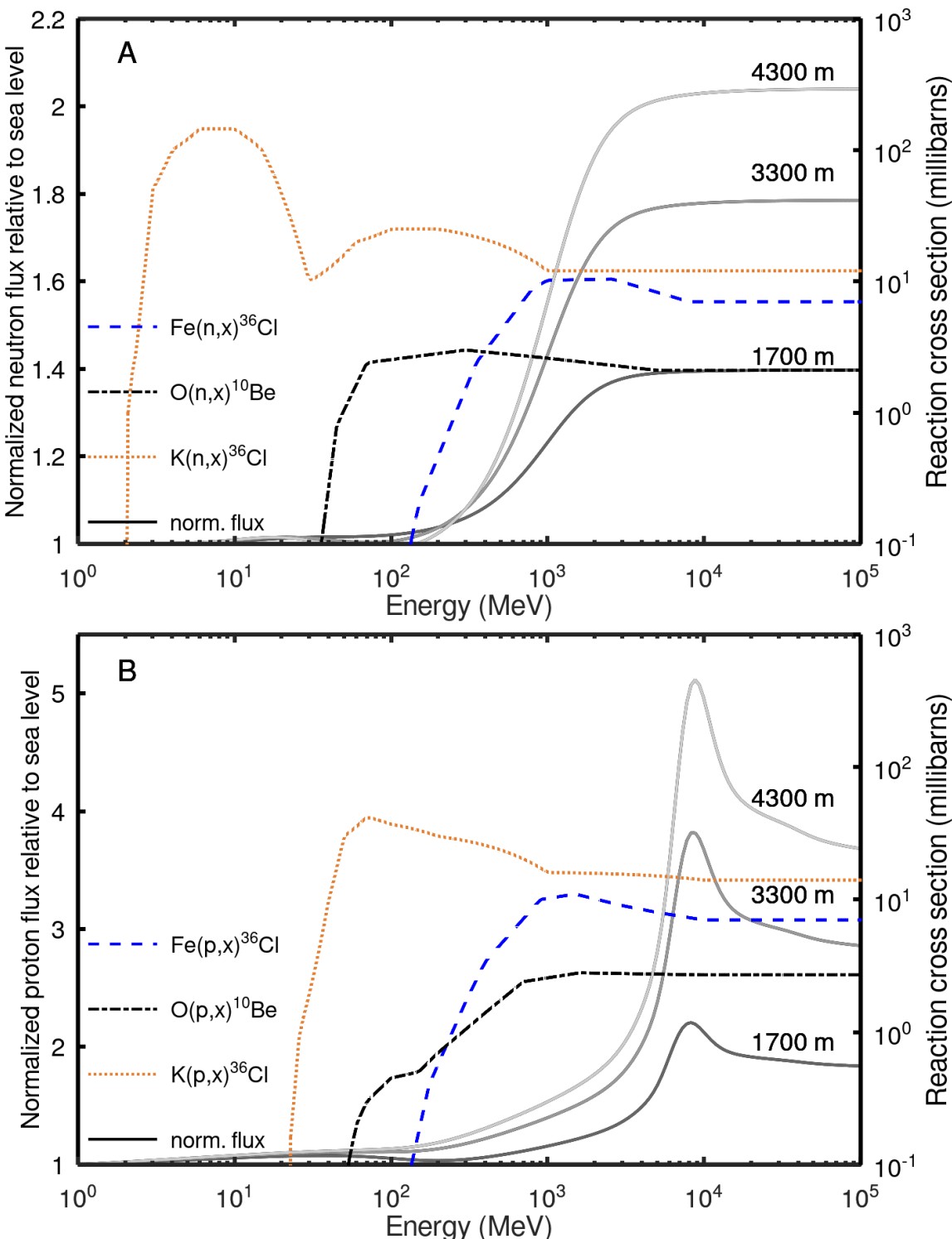

**Figure 1. Comparison of high-energy particle flux energy spectra (grey curves, left y-axis) with excitation functions (dashed curves, right y-axis). A. Neutron energy spectra at 1700 m, 3300 m, and 4300 m asl (corresponding to the elevations of the calibration sites) at a cutoff rigidity of 6.3 GV and a long-term solar modulation constant of 462 MV (the approximate average at the calibration sites since 21 ka, the last glacial maximum) normalized by the flux at 1 MeV to account for increased neutron fluxes with altitude and by the spectrum at sea level to highlight changes in the energy spectrum with altitude (left y-axis) and excitation functions for spallation production of $^{36}$Cl from Fe and K and $^{10}$Be from O (Reedy et al., 2013) (right y-axis). B. Equivalent normalized proton energy spectra and excitation functions.**

120

**Table 1. Sample locations and characteristics**

| ID | Latitude dec. deg. | Longitude dec. deg. | Elev. m asl | Thickness[1] cm | Feature | Exposure Model |
|---|---|---|---|---|---|---|
| *Owens Valley* | | | | | | |
| OV19-1 | 37.416 | -118.630 | 1702 | 12 | moraine | constant exposure |
| OV19-2 | 37.416 | -118.630 | 1701 | 12 | moraine | constant exposure |
| OV19-3 | 37.416 | -118.630 | 1701 | 10 | moraine | constant exposure |
| *Sierra Nevada* | | | | | | |
| BL15-1 | 37.166 | -118.618 | 3365 | 3 | erratic | constant exposure |
| BL15-2 | 37.167 | -118.619 | 3363 | 4 | erratic | constant exposure |
| BL15-3 | 37.167 | -118.622 | 3360 | 4 | erratic | constant exposure |
| LL15-1 | 37.392 | -118.766 | 3328 | 6 | erratic | constant exposure |
| LL15-2 | 37.403 | -118.756 | 3318 | 4 | erratic | constant exposure |
| *Mt. Evans* | | | | | | |
| 97EV14B | 39.588 | -105.644 | 4267 | 25 | blockfield | steady-state erosion |

[1]**All samples are granitic in composition with an assumed density of 2.7 g cm$^{-3}$.**

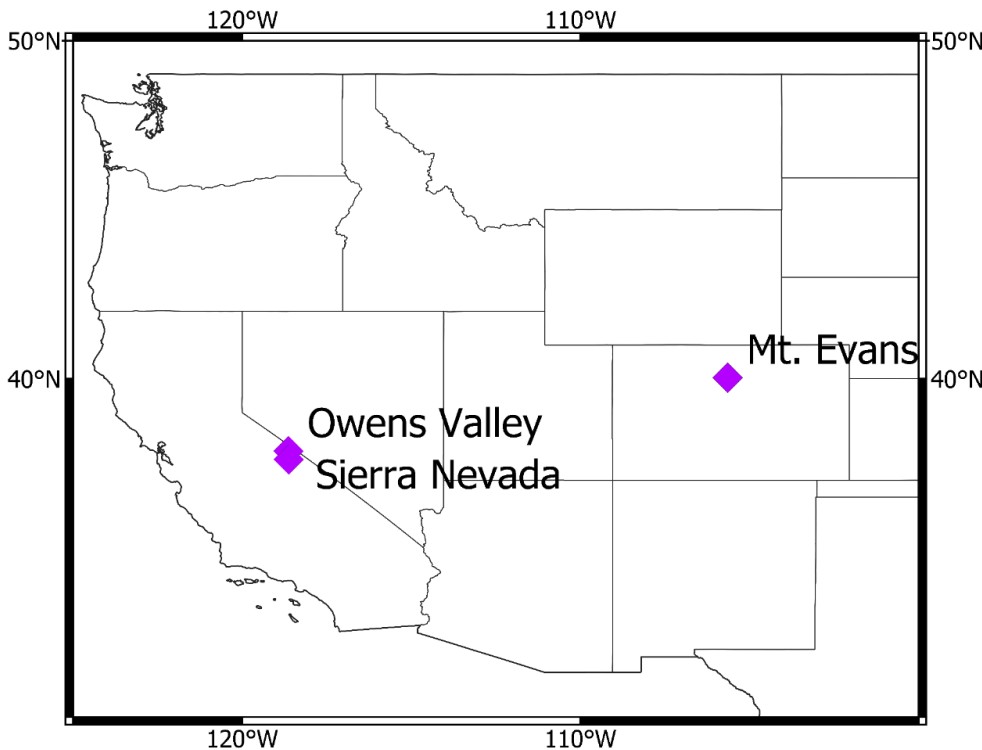

**Figure 2. Location of the calibration sites within western North America (purple diamonds). The sites are located between 38-40°N**
 **minimizing inter-site variability in cutoff rigidity.**

### 2.3 Analytical methods

The calibration samples were crushed, and magnetite and K-feldspar were separated and prepared for $^{36}$Cl measurement. Magnetite was isolated from the crushed rock using repeated cycles of magnetic separation with rare-earth magnets and grinding with zirconia balls on a wrist-action shaker. Separates were leached once in dithionite-citrate-bicarbonate to remove secondary Fe-oxide minerals and then in 10% nitric acid followed by 1% NH$_4$(OH) to remove any adsorbed chloride. Clean magnetite was spiked with a $^{35}$Cl enriched carrier ($^{35}$Cl/$^{37}$Cl = 273) and dissolved in high purity oxalic acid. After dissolution, precipitates were removed

by centrifugation. Next, Cl was precipitated from solution as AgCl by adding $HNO_3$ and an excess of $AgNO_3$. Feldspar separates were obtained from the 250-500 μm fraction of the crushed sample through froth flotation and density separation in lithium heteropolytungstate. Feldspar grain surfaces were cleaned in 10% $HNO_3$. The samples were then spiked and dissolved in a solution of ca. 30% HF and 1% $HNO_3$. After dissolution, solutions were refrigerated to 2 °C to promote formation of fluoride precipitates, which were removed by centrifugation to reduce solution viscosity. Chloride was then precipitated from solution as AgCl. The AgCl precipitates from both the magnetite and feldspar samples were dissolved in $NH_4(OH)$ and purified of $SO_4^{2-}$ through precipitation with $Ba^{2+}$ and anion chromatography. The final AgCl product was dried and loaded into AgBr-filled copper cathodes for measurement of $^{36}Cl/Cl$ and $^{35}Cl/^{37}Cl$ by accelerator mass spectrometry (AMS) at PRIME Lab, Purdue University. Measurements were normalized to the Sharma et al. (1990) standard, using the dilution with a $^{36}Cl/Cl$ ratio of 1600 x $10^{-15}$.

Sample target and bulk-rock chemistries, necessary for modeling $^{36}Cl$ production, were determined by inductively coupled plasma-optical emission spectrometry (ICP-OES) using a Horiba *Ultima Expert™* spectrometer. Aliquots of sample material were taken for target chemistry measurements immediately prior to dissolution. Approximately 50 mg was dissolved in concentrated HF and then evaporated to dryness, redissolved in $HNO_3$, and again evaporated to dryness before being taken-up in 5% $HNO_3$ for measurement. Stable Cl concentrations were determined from isotope dilution of the $^{35}Cl$-enriched carrier. For measurement of trace elements important for controlling low-energy neutron fluxes in the subsurface (i.e., B, Sm, U, Th, Gd, Cr, and Li), ca. 20 g of rock was powdered in a ring-and-puck mill and ca. 250 mg of homogenized powder dissolved in HF and prepared for measurement in the same way as the target fraction. However, this approach cannot be used to determine Si, which forms a volatile fluoride. To determine Si and other major element concentrations, ca. 100 mg of rock powder was fused in a 50:50 mixture of lithium-metaborate and lithium-tetraborate at 1000°C for 30 minutes. The fusion cake was dissolved in 50% $HNO_3$ and diluted to 5% $HNO_3$ for measurement. Analytical water was approximated from loss on ignition (LOI) at 1000° C, assuming all evolved volatiles represent $H_2O$.

Quartz separates were obtained from the 250-500 μm fraction of the crushed samples using froth flotation and leaching in 1% HF/$HNO_3$. The quartz separates were spiked with an in-house $^9Be$ carrier and dissolved in concentrated HF. After dissolution, 1 ml $H_2SO_4$ was added to maintain volume and then the HF was evaporated. Beryllium was isolated from the $H_2SO_4$ using a rapid separation scheme. First, amphoteric species were partitioned into the supernatant by precipitation of other species at pH 14. Beryllium was then precipitated from the supernatant by adjusting to pH 9 with HCl, which coprecipitates some Al and Fe. The precipitates were dissolved in 0.4 M oxalic acid and negatively charged oxalate complexes of Al and Fe were separated from the neutral $Be(C_2O_4)^0$ complex by passing the solution through an anion exchange column (Dowex 1x8 100-200 mesh). Beryllium was precipitated directly from the oxalic acid solution with $NH_4(OH)$ and polished by reprecipitating twice. The $Be(OH)_2$ product was calcined under an acetylene flame, mixed with Nb binder, and loaded into a stainless steel cathode for $^{10}Be/^9Be$ measurement by AMS. Measurements were normalized to the standard described by Nishiizumi et al., (2007) with a nominal $^{10}Be/^9Be$ ratio of 2.851 x $10^{-12}$.

### 2.4 Calculations

#### 2.4.1 Production rate calibration

Production rates of $^{36}Cl$ from Fe are calibrated against $^{36}Cl$ in K-feldspar and $^{10}Be$ in quartz. The calibrations assume two end-member geomorphic scenarios: constant exposure and no erosion for the Owens Valley samples, which are moraine clasts, and steady-state erosion for the Mt. Evans sample, which was collected from an unglaciated alpine blockfield. The steady-state erosion assumption for this sample is supported by its position adjacent to the steady-state erosion line on a $^{36}Cl/^{10}Be$ two-isotope plot (Figure 4). In the constant exposure model, a site-specific production rate of $^{36}Cl$ from Fe in magnetite is calculated from the ratio of the $^{36}Cl$ concentration in magnetite to feldspar, multiplied by the total production rate in feldspar, minus production from

pathways other than spallation from Fe. This is then normalized by the Fe concentration in magnetite to derive the [36]Cl production rate from Fe at the site:

$$P_{36,Fe} = \left( \frac{N_{36,mt}}{N_{36,fs}} \sum_{i=K,Ca,Ti,Fe,Cl} P_{36,i} [i]_{fs} - \sum_{i=K,Ca,Ti,Cl} P_{36,i}[i]_{mt} \right) / [Fe]_{mt} \qquad \text{Eqn. 2}$$

where $N_{36, mt}$ and $N_{36, fs}$ are the concentrations of cosmogenic [36]Cl in magnetite (mt) and feldspar (fs) (atoms g$^{-1}$), $P_{36, i}$ is the production rate of [36]Cl on target element i at the study site (atoms g$^{-1}$ yr$^{-1}$), and $[i]_{fs}$ and $[i]_{mt}$ are the concentrations of target element i in the feldspar and magnetite separates, respectively.

In the steady-state erosion scenario production is integrated across the exhumation path such that the attenuation length-scale of each reaction in the subsurface is also important. To calibrate the site-specific production rate of [36]Cl from Fe, the erosion
rate (E) (g cm$^{-2}$ yr$^{-1}$) is first determined by implicitly solving Eqn. 3 using the [36]Cl concentration measured in feldspar:

$$N_{36} = \sum_i \sum_j \frac{P_{36,i,j}[i]}{\lambda + \frac{E}{\Lambda_j}} \qquad \text{Eqn. 3}$$

where $\Lambda_j$ is the attenuation length (g cm$^{-2}$) of the cosmic radiation responsible for production mechanism j and $\lambda$ is the decay constant of [36]Cl (t$^{-1}$), and $N_{36}$ is the concentration of [36]Cl (atoms g$^{-1}$). The erosion rate is then used in conjuncture with $N_{36,mt}$, and production rates from pathways other than spallation on Fe to again solve Eqn. 3, but this time for the site-specific production rate
of [36]Cl from Fe. Importantly, Eqn. 3 gives more weight to production from muon interactions than Eqn. 2 because of the longer muon than nucleon attenuation length in the subsurface.

Production of [36]Cl from Fe is also calibrated relative to [10]Be in quartz, which requires a formulation that accounts for differential decay between [10]Be and [36]Cl. Here, the approaches discussed in Moore and Granger (2019a) for the constant exposure and steady-state erosion cases are used. Calibrated production rates are then corrected for [36]Cl production pathways in magnetite
other than spallation from Fe and normalized by the Fe concentration in the same manner as in Eqn. 2. For the mid-elevation Sierra Nevada sites, the production rates calibrated in Moore and Granger (2019b) are adopted without modification (Table 2).

Reference spallation production rates at SLHL are taken from Borchers et al. (2016) and are scaled to the study sites using reaction-specific scaling factors from the Lifton-Sato-Dunai model (Lifton et al., 2014). Muon production of [36]Cl and [10]Be is modeled with depth in the subsurface following Marrero et al. (2016a) and parameterized by fitting coefficients and attenuation
lengths for two exponential terms to the slow-negative muon depth profile and a single exponential term to the fast muon depth profile. This approach captures the altitude dependence of muon attenuation lengths (Balco, 2017).

**2.4.2 Reaction-specific spallation attenuation lengths**

Spallation reactions that display different attenuation behavior in the atmosphere likely also differ in the subsurface. Capturing this effect is important for accurately calibrating production rates using samples from eroding surfaces. The higher
average atomic mass of nuclei in the subsurface than the atmosphere leads to a higher nucleon multiplicity (i.e., the average number of nucleons ejected from a target nucleus during a cosmic-ray reaction). Reactions that are sensitive to the energies of tertiary nucleons, such as production of [36]Cl from K, which has a threshold of about 1 MeV, should therefore attenuate more slowly in the subsurface than other reactions (Argento et al., 2015b).

This effect was explored using Monte Carlo modeling by Argento et al. (2015b) who presented polynomial functions
describing production of [36]Cl from K with depth in several rock types. To re-parameterize Argento et al.'s (2015b) results in a form conducive to use in Eqn. 3, we fit exponential functions to the polynomials for [36]Cl from K and [10]Be in quartz production with depth in granite. However, because the attenuation length for the fit to the [10]Be in quartz polynomial is lower (141 g cm$^{-2}$) than the conventional value for the [10]Be in quartz attenuation length in rock (ca. 160 g cm$^{-2}$), rather than using the best-fit [36]Cl attenuation length directly, we multiply the modeled [36]Cl$_K$/[10]Be$_{qtz}$ attenuation length ratio (ca. 1.1) by 160 g cm$^{-2}$ to estimate the attenuation

length for $^{36}$Cl from K production in rock. Although this approach does not capture the precise shape of the modeled $^{36}$Cl from K production profile, the accuracy is likely sufficient when considering only depth-integrated production under steady-state erosion.

To model subsurface attenuation for $^{36}$Cl production from Fe, which has too high of a threshold energy to be affected by increases in nucleon multiplicity in the subsurface, we use the atmospheric production model described in section 2.1. This approach assumes that the evolution of the cosmic ray energy spectrum with depth in the subsurface, above the relevant threshold energies, is similar to that in the atmosphere. Production profiles for $^{36}$Cl from Fe and $^{10}$Be in quartz were generated between the elevation of Mt. Evans (4300 m) and sea level, equivalent to ca. 410 g cm$^{-2}$ of mass-depth (the model is not valid below sea level) and exponential functions were fit to each. Attenuation lengths for all reactions are longer in the subsurface than in the atmosphere because the average atomic mass in rock is greater than in air. Thus, the modeled attenuation lengths in the atmosphere must be adjusted upwards. To do this, the modeled $^{36}$Cl$_{Fe}$/$^{10}$Be$_{qtz}$ attenuation length ratio in the atmosphere (ca. 0.9) is multiplied by 160 g cm$^{-2}$ to obtain the $^{36}$Cl from Fe subsurface attenuation length.

This approach gives unique attenuation lengths for each of the three relevant spallation reactions in the sample from Mt. Evans ($^{36}$Cl$_K$ = 176 g cm$^{-2}$, $^{36}$Cl$_{Fe}$ = 146 g cm$^{-2}$, $^{10}$Be$_{qtz}$ = 160 g cm$^{-2}$). Production of $^{36}$Cl from Fe thus declines more rapidly with increasing mass-depth in the subsurface than $^{10}$Be in quartz and much more rapidly than $^{36}$Cl from K. This implies that, when considering only spallation, a sample experiencing steady-state erosion at high elevation should have a $^{36}$Cl$_{Fe}$/$^{36}$Cl$_K$ concentration ratio that differs by ca. 20% from the ratio expected under constant exposure and no erosion.

### 2.4.3 Low-energy neutrons

Production of $^{36}$Cl by low-energy neutron capture by $^{35}$Cl is strongly influenced by snow cover and pore water because of the neutron scattering and absorbing properties of H. Modest snow cover increases the neutron flux just below the air/ground interface by preventing diffusive neutron leakage into the atmosphere (Zweck et al., 2013). Likewise, the presence of up to ca. 4 wt. % water in soil pores increases thermal neutron fluxes by enhancing thermalization in the subsurface, although larger amounts of pore water decrease fluxes by enhancing neutron absorption (Phillips et al., 2001). The Owens Valley samples are clasts that were embedded in the surface of a moraine and are subject to both the effects of snow shielding and of pore water in the adjacent morainal sediment. However, because low-energy neutron fluxes do not affect accumulation of $^{10}$Be in quartz, and because the production rate of $^{36}$Cl from K is known, the K-feldspar/quartz pair can be used to estimate sample-specific neutron capture production rates. We use these adjusted values for our production rate and ratio calibrations for the Owen's Valley site. A similar calculation is not necessary at Mt. Evans because of the low chloride concentrations.

### 2.4.4 Scaling factor ratios

To examine changes in $^{36}$Cl$_{Fe}$/$^{10}$Be$_{qtz}$ and $^{36}$Cl$_{Fe}$/$^{36}$Cl$_K$ with altitude, the calibrated production rates at the study sites are first normalized by the production rate of either $^{10}$Be in quartz or $^{36}$Cl from K at the samples' locations, whichever the calibration was conducted against, giving production ratios. To normalize differences in geomagnetic scaling between the sites, the production ratios at the calibration sites are then divided by the production ratios at sea level at their geographic locations. In this normalization scheme, all scaling factors are calculated using the reaction-specific scaling model of Lifton et al. (2014). Reference SLHL production rates are taken from Marrero et al. (2016a) for $^{10}$Be in quartz and $^{36}$Cl from K. The SLHL production rate of Fe is taken from the inverse-error weighted average SLHL production rate of the calibration samples (Table 2), excluding OV19-1 as an outlier (section 3.2). The total normalization can be expressed as:

$$R_{Fe/K,Be} = \left(\frac{P_{Fe,cal}}{S_{K,Be}*P_{K,Be,SLHL}}\right) \Big/ \left(\frac{S_{Fe,0}*P_{Fe,SLHL}}{S_{K,Be,0}*P_{K,Be,SLHL}}\right) = \left(\frac{P_{Fe,cal}}{S_{K,Be}}\right) \Big/ \left(\frac{S_{Fe,0}*P_{Fe,SLHL}}{S_{K,Be,0}}\right) \qquad \text{Eqn. 4}$$

where $R_{Fe/K,Be}$ is the resulting altitude scaling factor ratio, $P_{Fe,cal}$ is the production rate of $^{36}$Cl from Fe at the calibration site (as calculated from the data using the approach outlined in section 2.4.1), $S_{K,Be}$ is the scaling factor for production of $^{36}$Cl from K or $^{10}$Be in quartz from SLHL to the site, $S_{K,Be,0}$ is the scaling factor for production of $^{36}$Cl from K or $^{10}$Be in quartz from SLHL to sea

level at the site, and $S_{Fe,0}$ is the scaling factor for $^{36}Cl$ production from Fe from SLHL to sea level at the site. Finally, $P_{Fe,SLHL}$ is the production rate of $^{36}Cl$ from Fe at SLHL, which is determined from the average of all calibration data. The SLHL production rate of $^{36}Cl_K$ or $^{10}Be_{qtz}$ ($P_{K,Be,SLHL}$) is in both the numerator and denominator and thus cancels. This approach allows us to examine whether deviations in production ratios from the average ratio are consistent with the modeled scaling behavior by comparing calibrated scaling factor ratios with model predictions using statistical hypothesis testing and goodness-of-fit metrics.

## 3. Results

### 3.1 Modeling results

The modeled changes in $^{36}Cl_{Fe}/^{10}Be_{qtz}$ and $^{36}Cl_{Fe}/^{36}Cl_K$ production ratios with altitude are shown in Figure 3. The model predicts that the scaling of $^{36}Cl$ production from Fe departs significantly from lower energy reactions, increasing by approximately 40% from sea level to 5.5 km (ca. 500 hPa) relative to $^{10}Be$ in quartz and by slightly more relative to $^{36}Cl$ from K, which is sensitive to even lower energies (Figure 1). At the study sites, the model predicts an increase in the $^{36}Cl_{Fe}/^{10}Be_{qtz}$ production ratio of 18% and in the $^{36}Cl_{Fe}/^{36}Cl_K$ ratio of 22% between Owens Valley and Mt. Evans. Next, we test these predictions against empirical data.

### 3.2 Measurement results and scaling factor ratios

Measured $^{36}Cl$ concentrations in magnetite and K-feldspar, $^{10}Be$ concentrations in quartz, and target and bulk rock chemistries are presented in the supplement. All measured nuclide concentrations are corrected by procedural blanks, which account for less than 5% of the $^{36}Cl$ and 2% of the $^{10}Be$ inventory in any sample. Stable Cl inventories are corrected by blanks ranging from 73 to 100 µg of Cl. For the Mt. Evans magnetite sample, the blank correction produces a negative Cl concentration, so the Mt. Evans magnetite is assumed to have no stable Cl. Magnetite target chemistry is dominated by Fe, although low-energy neutron capture on Cl is also a significant production pathway in the Owens Valley samples. Spallation from K is the largest source of $^{36}Cl$ in all feldspar samples, while low-energy neutron capture is the second most important pathway (excepting in OV19-3, in which the two are subequal) (Supp. Table "Computed Parameters"). The radiogenic $^{36}Cl$ inventory is computed by assuming secular equilibrium between radiogenic production and decay (Marrero et al. 2016a) and subtracted from the reported concentrations. These corrected concentrations are then used to calibrate production rates. Analysis of the effects of environmental water on the low-energy neutron fluxes from the K-feldspar/quartz pair at Owens Valley (section 2.4.3) indicates that neutron-capture production rates are 19-36% higher than in the absence of pore water and snow cover. These elevated neutron-capture production rates are used in the calibration. Calibrated production rates are shown in Table 2.

Calibrated scaling factor ratios for the Owens Valley samples range from $0.47 \pm 0.10$ to $1.15 \pm 0.08$ with inverse-error weighted means of $0.96 \pm 0.05$ for $^{36}Cl_{Fe}/^{36}Cl_K$ and $0.97 \pm 0.05$ for $^{36}Cl_K/^{10}Be_{qtz}$. In both cases, OV19-1 produces a scaling factor ratio that is $>3\sigma$ lower than the mean, which may be because of an overestimation of the stable Cl concentration in the magnetite and consequently of the importance of neutron capture. The blank correction accounts for more than 50% of the total Cl in this sample, suggesting that variability in the amount of Cl introduced during sample preparation between the blank and the sample may significantly impact the total Cl inventory. Therefore, we exclude this sample as an outlier, and recalculate the means as $1.13 \pm 0.06$ and $1.12 \pm 0.05$ for the $^{36}Cl_K$ and $^{10}Be_{qtz}$ normalized ratios, respectively. At Mt. Evans, the calibrated scaling factor ratios are $1.38 \pm 0.10$ for $^{36}Cl_{Fe}/^{36}Cl_K$ and $1.29 \pm 0.09$ for $^{36}Cl_{Fe}/^{10}Be_{qtz}$. Ratios generally increase with altitude (Figure 5).

### 3.3 Model evaluation

To evaluate the altitude scaling of $^{36}Cl_{Fe}$ production, we compare modeled scaling factor ratios to the calibrated values. First, we examine the null hypothesis that ratios at Mt. Evans (ca. 4300 m) and Owens Valley (ca. 1700 m) are identical. For the

$^{36}Cl_{Fe}/^{36}Cl_K$ ratio, the z-statistic is 2.23 and for the $^{36}Cl_{Fe}/^{10}Be_{qtz}$ ratio the z-statistic is 1.64, indicating probabilities of 0.01 and 0.05 for the null hypothesis and that therefore the high and low elevation sites are statistically distinguishable at greater than the 90% confidence level.

Next, we use the chi-square goodness-of-fit test to evaluate how well the reaction-specific model fits the calibration data. The chi-squared statistic measures the difference between modeled and observed values, where a lower value of the statistic implies

a tighter fit of the model to the observations (Bevington and Robinson, 1992). We also report p values for the hypothesis that the model describes the data and the chi-squared statistic normalized by the number of degrees of freedom (i.e., the reduced chi-squared statistic or MSWD). To compare only independent data, we calculate statistics for the $^{36}Cl_K$ and $^{10}Be_{qtz}$ normalized ratios independently. The reaction-specific model fits the mean $^{36}Cl_{Fe}/^{36}Cl_K$ ratios with a chi-squared statistic of 0.021 with 1 degree of freedom (MSWD = 0.021, p = 0.89) and the $^{36}Cl_{Fe}/^{10}Be_{qtz}$ ratios with a chi-squared statistic of 0.249 (2 degrees of freedom, MSWD

= 0.125, p = 0.88). Conversely, an integral flux scaling model that uses the same scaling factors for all reactions (i.e., a vertical line in Figure 5) fits the $^{36}Cl_{Fe}/^{36}Cl_K$ production ratio with a chi-squared statistic of 4.97 (1 degree of freedom, MSWD = 4.97, p = 0.03) and the $^{36}Cl_{Fe}/^{10}Be_{qtz}$ ratios with a chi-squared statistic of 3.37 (2 degrees of freedom, MSWD=1.69, p = 0.19). Thus, the uniform scaling factor model can be rejected at an α value of 0.1 for $^{36}Cl_{Fe}/^{36}Cl_K$ but cannot be rejected at this level for $^{36}Cl_{Fe}/^{10}Be_{qtz}$. However, in both cases, the reaction-specific scaling model fits the data more closely than the integral flux model. This may

indicate that the uncertainties on the empirical ratios are overestimated.

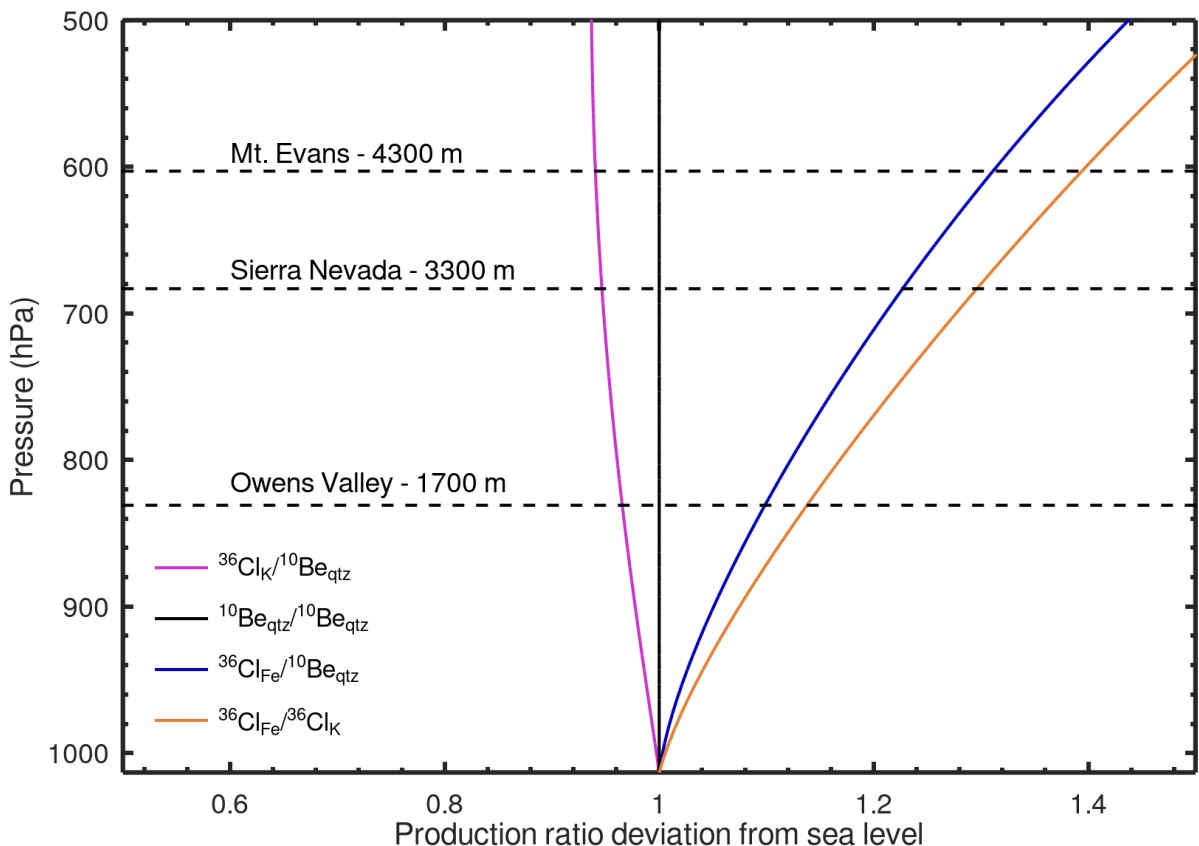

**Figure 3. Modeled deviation of cosmogenic nuclide production ratios from the ratio at sea level (i.e., 1013.25 hPa) at high latitude (cutoff rigidity = 0 GV, solar modulation constant = 462 MV) with increasing altitude (decreasing pressure). Curves are normalized to the curve**
**for $^{10}Be_{qtz}$ production. Production rates by lower energy spallation reactions (e.g., $^{36}Cl_K$) decrease modestly relative to $^{10}Be_{qtz}$ with increasing altitude (purple line). Conversely, production of $^{36}Cl_{Fe}$ increases with altitude relative to $^{10}Be_{qtz}$ (blue line), by more than 40%**

between sea level and 500 hPa (ca. 5.5 km asl). Dashed, horizontal lines indicate air pressure at the Owens Valley, Sierra Nevada, and Mt. Evans calibration sites estimated from the ERA40 model (Uppala et al., 2005) as implemented in Lifton et al. (2014).

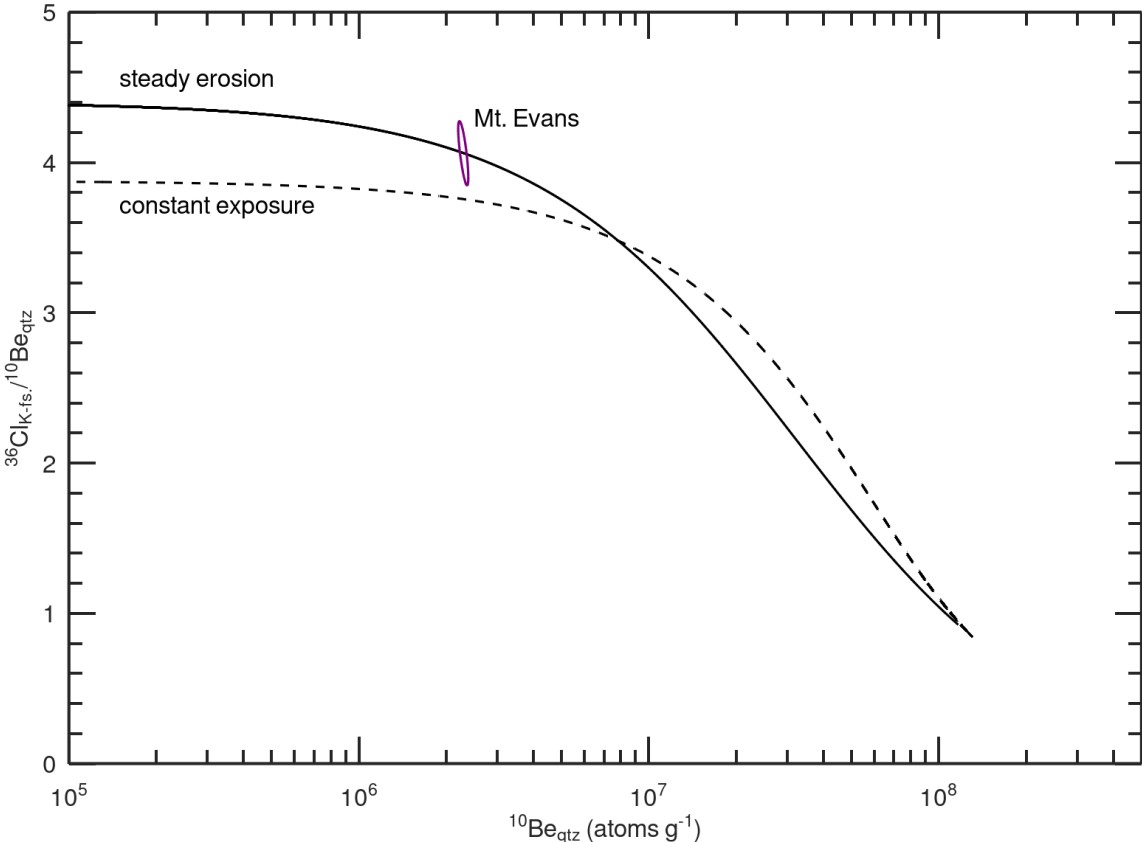

**Figure 4.** $^{36}Cl_{K\text{-fs.}}$ / $^{10}Be_{qtz}$ **two-isotope plot for the Mt. Evans sample. Curves are calculated using the sample-specific $^{36}$Cl production rate in K-feldspar (considering spallation, neutron capture, and muon reactions) and reaction-specific attenuation lengths (section 2.4.2). The error ellipse represents 2σ analytical errors on the $^{10}$Be concentration in quartz and $^{36}$Cl concentration in K-feldspar. The steady erosion line is elevated above the constant exposure line at low $^{10}$Be concentration (i.e., short exposure ages or fast erosion rates) because of the greater subsurface attenuation length of $^{36}$Cl production from K. The Mt. Evans sample overlaps with the steady erosion line at 1σ and is greater than 2σ from the constant exposure line, indicating that steady-state erosion is a more plausible geomorphic model than constant exposure.**

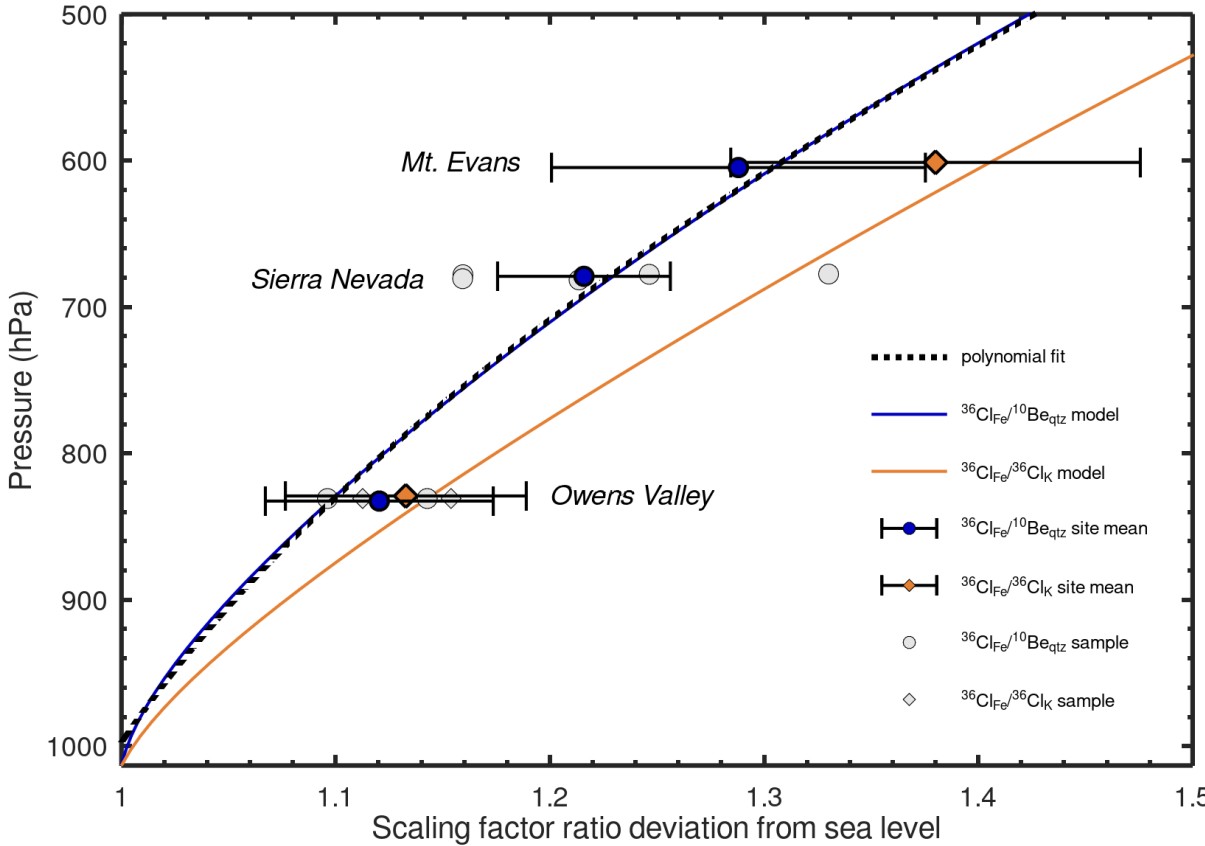

**Figure 5. Model/data comparison.** Modeled production ratio deviation from sea level for $^{36}Cl_K/^{10}Be_{qtz}$ (blue) and $^{36}Cl_{Fe}/^{36}Cl_K$ (orange). Data points show measured $^{36}Cl_{Fe}/^{36}Cl_K$ and $^{36}Cl_{Fe}/^{10}Be_{qtz}$ scaling factor ratios at the three calibration sites (OV19-1 has been omitted as an outlier). Site means are shown by blue- and orange-colored symbols and are offset in the y-direction to avoid error bar overlap. Error bars are 1σ. Measured production ratios show a general trend of increasing with decreasing air pressure. The ratios at the low and high elevation sites are distinguishable from each other for both reaction pairs at greater than the 90% confidence interval. The dashed, black line shows the second-degree polynomial fit to the modeled $^{36}Cl_{Fe}/^{10}Be_{qtz}$ ratios.

**Table 2. Calibrated $^{36}Cl$ production rates from Fe and $^{36}Cl_{Fe}/^{36}Cl_K$ and $^{36}Cl_{Fe}/^{10}Be_{qtz}$ scaling factor ratios**

| | Production rate from Fe at site | | Production rate from Fe at SLHL | | Scaling factor ratio | |
| | cal. vs. K-fs. | cal. vs. qtz. | cal. vs. K-fs. | cal. vs. qtz. | norm. to $^{36}Cl_K$ | norm. to $^{10}Be_{qtz}$ |
| ID | at. g Fe$^{-1}$ yr$^{-1}$ | at. g Fe$^{-1}$ yr$^{-1}$ | at. g Fe$^{-1}$ yr$^{-1}$ | at. g Fe$^{-1}$ yr$^{-1}$ | - | - |
|---|---|---|---|---|---|---|
| *New Data (this study)* | | | | | | |
| OV19-1 | 2.19±0.45 | 2.30±0.47 | 0.54±0.11 | 0.57±0.12 | 0.47±0.10[1] | 0.47±0.10 |
| OV19-2 | 5.21±0.37 | 5.34±0.37 | 1.29±0.09 | 1.32±0.09 | 1.11±0.08 | 1.10±0.08 |
| OV19-3 | 5.43±0.38 | 5.59±0.36 | 1.34±0.09 | 1.38±0.09 | 1.15±0.08 | 1.14±0.07 |
| 97EV14B | 36.9±2.6 | 36.8±2.5 | 1.31±0.09 | 1.31±0.09 | 1.38±0.10 | 1.29±0.09 |
| *Old Data (Moore & Granger, 2019)* | | | | | | |
| BL15-1 | - | 19.3±1.4 | - | 1.43±0.10 | - | 1.32±0.09 |
| BL15-2 | - | 18.1±1.4 | - | 1.34±0.10 | - | 1.24±0.09 |
| BL15-3 | - | 16.8±1.3 | - | 1.25±0.10 | - | 1.15±0.09 |
| LL15-1 | - | 16.6±1.1 | - | 1.25±0.08 | - | 1.15±0.08 |
| LL15-2 | - | 17.3±1.4 | - | 1.31±0.10 | - | 1.20±0.10 |

[1] OV19-1 omitted from the analysis as an outlier (see section 3.2)

## 4. Discussion

### 4.1 Implications

These results imply that $^{36}$Cl production from Fe can be more accurately scaled using reaction-specific scaling factors than integral-flux scaling factors. This is significant for assessing $^{36}$Cl production rates in Fe-oxide minerals, such as magnetite, which is a promising target mineral for determining catchment-averaged erosion rates on mafic rocks (Moore and Granger, 2019a). When normalized to $^{10}$Be in quartz, the reaction-specific scaling model using the new calibration data produces an inverse-error weighted SLHL reference production rate of $1.35 \pm 0.06$ atoms g Fe$^{-1}$ yr$^{-1}$ and $1.31 \pm 0.07$ atoms g Fe$^{-1}$ yr$^{-1}$ when normalized to $^{36}$Cl in K-feldspar. Both reference production rates are within $2\sigma$ error of the estimate of $1.28 \pm 0.03$ atoms g Fe$^{-1}$ yr$^{-1}$ made by Moore and Granger (2019b).

This work also suggests the possibility of applications of $^{36}$Cl in different minerals that exploit reaction-specific scaling or differences in subsurface attenuation lengths. For example, for a sample that is known to have experienced steady-state erosion or constant exposure, the $^{36}$Cl concentration ratio between magnetite and K-feldspar might be inverted for the exposure altitude. Given a 5% error in the $^{36}$Cl ratio determination, the model indicates that the exposure altitude could be resolved to ca. $\pm 600$ m ($1\sigma$). Likewise, if, as argued in section 2.4.2, there is a ca. 20% difference in the subsurface attenuation length between production of $^{36}$Cl from Fe and from K by spallation, then this should lead to a difference in $^{36}$Cl ratios between Fe-rich and K-rich mineral separates that could measurably differentiate between steady-state erosion and constant exposure at high erosion rates and short exposure ages (Figure 4). Further work is needed to empirically evaluate the subsurface attenuation lengths of these reactions.

### 4.2 Polynomial parameterization

The increase in the scaling factor for $^{36}$Cl production from Fe with increasing altitude relative to a uniform scaling factor model, such as Lal/Stone or the integral flux version of Lifton/Sato/Dunai (Lifton et al., 2014; Stone, 2000), can be reasonably approximated using a second degree polynomial:

$$P_{36,Fe}(z, \lambda) = P_{36,Fe}(SLHL) * S(z, \lambda) * [a(1013.25hPa - z)^2 + b(1013.25hPa - z) + c] \qquad \text{Eqn. 5}$$

Where $P_{36,Fe}(z, \lambda)$ is the scaled production rate of $^{36}$Cl from Fe at atmospheric pressure z (hPa) and cutoff rigidity $\lambda$ (GV), $P_{36,Fe}$(SLHL) is the reference production rate of $^{36}$Cl from Fe at SLHL, $S(z,\lambda)$ is the scaling factor from an integral flux scaling model, and a, b, and c are polynomial coefficients. This formulation allows a straightforward modification of integral flux scaling factors that are commonly used to scale $^{10}$Be production to accommodate $^{36}$Cl scaling from Fe that captures the shape of the curve in Figure 5 with good fidelity between sea level (1013 hPa) and ca. 5.5 km (500 hPa). Best-fit polynomial coefficients calculated at 1 GV intervals between 0 and 14 GV are presented in the supplement, although the shape of the curve is only weakly sensitive to cutoff rigidity.

## 5. Conclusions

Modeling using excitation functions and cosmic ray nucleon fluxes predicts that $^{36}$Cl production from Fe attenuates more rapidly in the atmosphere than reactions that are sensitive to lower energies. The $^{36}$Cl$_{Fe}$/$^{36}$Cl$_K$ production ratio is predicted to increase with elevation at an average rate of approximately 0.84% per 100 m between sea level and the elevation of the Mt. Evans sample (4300 m asl) and by slightly less when normalized to $^{10}$Be in quartz. We evaluated this prediction against measured scaling factor ratios between $^{36}$Cl in magnetite and $^{36}$Cl in K-feldspar and $^{10}$Be in quartz across an altitude transect. Samples with late-glacial exposure histories at elevations of ca. 1700 and 4300 m asl in western North America produce scaling factor ratios that differ at greater than the 90% confidence level and that fit a reaction-specific scaling model more closely than a uniform flux

scaling model. Thus, reaction-specific scaling factors are likely appropriate for scaling production of $^{36}$Cl from Fe, especially when analyzing $^{36}$Cl in magnetite and other Fe-rich materials where spallation from Fe is the dominant production pathway.

**Competing interests:**

The contact author has declared that none of the authors has any competing interests.

**Acknowledgements:**

This work was funded by NSF-EAR-2011342 to DEG. We would like to thank Will Odom and Zach Meyers for collecting the Owens Valley samples, the PRIME Lab staff for AMS measurements, Christopher Halsted and Irene Schimmelpfennig for reviews

that helped improve this manuscript, and Greg Balco for editorial handling.

**Code and data availability:**

All new research data is provided in the supplement. Codes used to calibrate production rates are available at https://github.com/magnesiowustite/Altitude-scaling-of-Cl-36-production-from-Fe.

**Author Contributions:**

AM conceptualized the study and conducted the investigation and formal analysis. AM and DG acquired funding and DG provided laboratory resources. AM wrote the manuscript with input from DG.

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
