# Peer review of "Technical Note: Altitude scaling of 36Cl production from Fe"

_EGUsphere, 2024_

## Author Response (AR1)

**We would like to thank Christopher Halsted for the very helpful review. We are pleased to implement the suggested reorganization as well as the minor and technical changes and improve the presentation of our statistical analysis. Our responses to individual reviewer points are recorded in purple text.**

This preprint is remarkably well-written for being in the early stages of review. As a result, I have few purely technical corrections. The model construction and sample analysis procedures are in line with established practices and I see no flaws in their execution. There is a part of me that wonders about what uncertainties you take on by not knowing the original geometry of the Mt. Evans sample, but I think your approach of estimating sample thickness from measured mass is entirely reasonable. I have two general pieces of constructive criticism for the authors to consider, the first relates to the statistical analysis of the model/sample comparison and the second relates to the overall manuscript organization.

On the statistics side, I would like for more information to be provided about the chi-square goodness of fit test procedures and your interpretations. I provide more detailed feedback in the "Specific Comments" section of this review, but I found that I was reviewing section 4.1 several times to make sure I understood your results and interpretation correctly, and more information would help in my understanding of the test parameters and to ensure that your interpretations are reasonable.

On the organizational side of things, I have some suggestions for the authors to consider. This is not technical feedback, so the choice to adopt or ignore these suggestions is completely up to the authors, but I think they will help with the readability of the manuscript. Currently, the methods and results of your modeling is in the Methods section, while the methods and results for your calibration of nuclide production rates and scaling factors is in the Data Analysis and Results section. This made it a bit difficult on first read to directly compare your model results to your empirical results. I think some re-organization to put the methods for both modeling and empirical calculations in one section, and the results of both approaches in the other section, would help with readability. More detailed suggestions are provided in the Specific Comments.

**Specific Comments**

Introduction

- In the final line of the introduction, you mention how your results have implications for cosmogenic studies using Fe-rich rocks. This feels like an important outcome of this work but is not mentioned before or after. I think it would be worth expanding on this in the introduction, explaining that

confident $^{36}$Cl analysis in Fe-rich rocks could be valuable, particularly in quartz-poor areas.

We have added the following text (Lines 61-64)

"The results have implications for accurately estimating $^{36}$Cl production rates in Fe-rich rocks (e.g., peridotite or basalt) and mineral separates (e.g., magnetite), which are increasingly used to determine erosion rates and exposure ages in quartz-poor mafic and ultramafic landscapes (e.g., Leontaritis et al., 2022; Moore et al., 2024; Moore and Granger, 2019a)."

Methods

- Section 2.1:
  - The final paragraph (lines 91 – 94) and Figure 2 are results, it might be worth creating a "Modeling Results" (or some other name) results section to house all of your results in one section, as readers might want to easily compare the model/empirical results.

We have created a "Modeling results" section (Section 3.1, lines 262-268).

- Section 2.2:
  - Is there any chance that substantial differences in time-integrated geomagnetic cutoff rigidities could exist between Mt. Evans and Owens Valley? If not, it would be worth explicitly stating this in section 2.2. X

The time integrated cutoff rigidity is slightly lower at Mt. Evans than at Owens Valley (i.e., ca. 4.9 GV vs. 6.3 GV). However, the scaling factor ratios are only weakly sensitive to cutoff rigidity, so this difference is not very significant.

Moreover, normalizing the production ratio at the calibration sites by the production ratio at sea level at the site's geographic locations should remove this difference. We have added a sentence explaining that this is the reason for this normalization:

"To normalize differences in geomagnetic scaling between the sites, the production ratios at the calibration sites are then divided by the production ratios at sea level at their geographic locations." (Lines 244-246)

- o Again, the last sentence here (lines 107-108) sounds like results and could be relegated to a dedicated results section that would allow for easy model/measurement comparisons

We have moved this to the results (Section 3.1).

- Section 2.3:
  - o I do not see information in here or in the supplement about the $^9$Be carrier concentration X

We report the amount of carrier added in terms of mass of beryllium, rather than mass of carrier solution, so the carrier concentration is not needed to recalculate the data. We note units of mg Be and mg Cl for the carrier masses in the supporting information.

Data Analysis and Results

- To be honest, I am a bit torn about the organization of this section. It is quite long, over 3 pages, and each subsection almost reads as its own mini paper, often including bits of what might be considered intro, background, methods, and results. While in some ways this works well, keeping the information about each aspect of the production rate calculation process contained in its own section, it did feel like I was jumping from topic to topic a bit as I was reading. This is not a technical critique, the information in all these sections seems sound, but more of an organizational consideration. If you are agreeable, one possible re-organizing to consider would be to take all the methods associated with production rate calculations (e.g., in sections 3.2, 3.3, 3.4, and 3.5) and bundle those together in a dedicated Methods section where they all fall under the heading of considerations for calculating and calibrating nuclide production rates and scaling factors. The results section can then be a lot shorter, simply stating (1) the results of your modeling efforts (which are now in the Methods section) and (2) the results of your sample analyses and calibrated nuclide-specific production rates and scaling factors. This would make the comparison of the two a bit easier to facilitate for the reader.

We have implemented this reorganization. We have created section 2.4 "Calculations" and moved all the information from 3.2-3.5 into this section.

- Table 2 needs a more informative caption. Specify that these are $^{36}Cl_{Fe}$ production rates and scaling factor ratios. Additionally, it might be worth indicating somehow that sample OV19-1 was omitted from calculated scaling factors, and/or indicate that in the Figure 5 caption.

We have added this information to the table caption and a footnote that OV19-1 was removed as an outlier with reference to section 3.2. We have also added that OV19-1 was omitted as an outlier in the caption to Figure 5.

Discussion

- Section 4.1:
    - I think a table to summarize your hypothesis tests and results would be a nice way to visualize this information

To maintain brevity, we would prefer not to add another table. In total, we performed 2 tests for 2 datasets, which we think we can adequately explain in the text.

- Some thoughts on the chi-square test procedures and interpretations (lines 311-317):
    - Some more explanation is needed about the chi-square goodness-of-fit test (perhaps also a reference to provide more information). I was a bit confused about this on my first read-through because you do not specify the null hypothesis for this test and do not specify how many degrees of freedom are present in each calculation (which is important for understanding how the MSWD values are calculated). Additionally, I at first interpreted this test as the calculation of a *reduced* chi-square statistic, which is commonly used in isotope geochemistry, and so I was extra confused when I saw reported MSWD values (aka, reduced chi-square statistic values) that sometimes agreed and sometimes did not agree with the reported chi-square values. Because of the lack of information, the MSWD and p-values confused me at first because I interpreted them as *rejecting the alternative hypothesis that the reaction-specific models are better fits than the integral flux scaling model*, which was contrary to your conclusions at the end of this paragraph. Additionally, because the degrees of freedom and calculation details are not provided, it took some time and inferring on my part to understand why the MSWD and chi-squared statistic values sometimes agreed and sometimes didn't.
    - To help clarify all of this for other readers, you should identify what the null hypothesis is for these tests and provide needed information including how many degrees of freedom are present for each ratio pair and what your alpha level is. Granted, it is entirely possible that I am misunderstanding what is being reported, but this is all the more

reason to expand your explanation for other readers and provide information about exactly which test(s) are being performed and how they should be interpreted.

- We have added brief explanations of the chi-squared test, the hypotheses that we test, the number of degrees of freedom, and a reference to classic data analysis text.

  - While on the topic of the chi-square tests, I would also like some more detail about how you are interpreting the reported MSWD values. The reported MSWD values suggest that the reaction-specific models are overfitting the data (they are very low), while the integral flux $Cl_{Fe}/Be_{qtz}$ scaling model actually has a decently good fit to the data (MSWD closest to 1), but you state in this section that "the data fit the reaction-specific scaling model more closely than the integral flux model". Again, it is possible I am misinterpreting the reported values, but more explanation about your own interpretations of these values would help clarify this.

This is true, although the p-values of 0.88 and 0.89 suggest that we cannot state that the reaction-specific model is overfitting the data at the 90% or 95% confidence level. In other words, the lower MSWD could just result from random error. The sample size is very small. It is also possible that the uncertainties in the scaling factor ratios are somewhat overestimated. Smaller errors would increase the MSWD for both models. This would move the reaction-specific model closer to 1 and the uniform-flux model further away.

The revised text reads (Lines 2297-309) reads:

"Next, we use the chi-square goodness-of-fit test to evaluate how well the reaction-specific model fits the calibration data. The chi-squared statistic measures the difference between modeled and observed values, where a lower value of the statistic implies a tighter fit of the model to the observations (Bevington and Robinson, 1992). We also report p values for the hypothesis that the model describes the data and the chi-squared statistic normalized by the number of degrees of freedom (i.e., the reduced chi-squared statistic or MSWD). To compare only independent data, we calculate statistics for the $^{36}Cl_K$ and $^{10}Be_{qtz}$ normalized ratios independently. The reaction-specific model fits the mean $^{36}Cl_{Fe}/^{36}Cl_K$ ratios with a chi-squared statistic of 0.021 with 1 degree of freedom (MSWD = 0.021, p = 0.89) and the $^{36}Cl_{Fe}/^{10}Be_{qtz}$ ratios with a chi-squared statistic of 0.249 (2 degrees of freedom, MSWD = 0.125, p = 0.88). Conversely, an integral flux scaling model that uses the same scaling factors for all reactions (i.e., a vertical line in Figure 5) fits the $^{36}Cl_{Fe}/^{36}Cl_K$ production ratio with a chi-squared statistic of 4.97 (1 degree of freedom, MSWD = 4.97, p = 0.03) and the $^{36}Cl_{Fe}/^{10}Be_{qtz}$ ratios with a chi-squared statistic of 3.37 (2 degrees of freedom, MSWD=1.69, p = 0.19). Thus, the uniform scaling factor model can be rejected at an α value of 0.1 for $^{36}Cl_{Fe}/^{36}Cl_K$ but cannot be rejected at this level for $^{36}Cl_{Fe}/^{10}Be_{qtz}$. However, in both cases, the reaction-specific

scaling model fits the data more closely than the integral flux model. This may suggest that the uncertainties are somewhat overestimated.**"**

- o Finally, if you decide to adopt my re-org suggestions and directly report the model scaling factor ratios and calibrated values together in the results section, Section 4.1 could also be moved to the results section, as it reads like what might be traditionally considered "results" rather than a discussion point.

We have moved section 4.1 to the results (now section 3.3, lines 292-309)

**Technical Corrections (indicated by preprint line number)**

32: Consider changing "show" to "suggest", as this is based on modeling and not empirical evidence

We have changed "show" to "suggest". (Line 32)

72-73: Should provide citations to show examples of irradiation experiments and cosmic ray cascade modeling

We have added some references here:

"Cross sections are typically derived from irradiation experiments (e.g., Schiekel et al., 1996), whereas particle fluxes are usually estimated from Monte Carlo modeling of the cosmic ray cascade (e.g., Masarik and Beer, 2009)." Lines 75-76

267: As written, "and low-levels of pore water by increasing neutron…" sounds like it needs a grammatical fix to improve readability. X

Changed to:

"Likewise, the presence of up to ca. 4 wt. % water in soil pores increases neutron fluxes by enhancing thermalization in the subsurface, although larger amounts tend to decrease fluxes by enhancing neutron absorption (Phillips et al., 2001)" Lines 233-235.

**Review 2:**

We would like to thank Irene Schimmelpfennig for the very useful comments on the manuscript and thorough review of the tables, figures, and supplement. We are pleased to incorporate these into our revised manuscript. Our responses to specific questions are recorded in purple text following the question.

In this paper, reaction-specific altitude scaling of 36Cl production rates from Fe is investigated by cross-calibrating the production rate of this reaction against those of 36Cl from K in feldspar and 10Be in quartz in the same samples at three different altitudes. I find the majority of the manuscript easy to read (except for a few parts, which I commented on below), the physics very well explained and the approaches well designed. The study is very useful in the light of using 36Cl analyses in magnetite.

I have arbitrarily checked the calculations of 2 samples from the AMS results to the 36Cl production contributions of the various reactions using my excel spreadsheet, and obtain broadly the same results (see one question about this below at the end). I don't have the expertise to evaluate the Polynomial parameterization.

I have one general question: As far as I understand, you scale the production rates of all reactions by default with the reaction-specific scaling factors of Lifton et al. (2014). By doing so, isn't there a circular component in your approach? Did you check the effect of using non-reaction-specific scaling factors?

This is an excellent question and one that warrants careful consideration. We believe that it is unlikely that the results would differ significantly using integral flux scaling factors because the reaction-specific effect for all reactions apart from spallation on Fe and Ti is quite small. For example, the reaction-specific $^{36}Cl_K/^{10}Be_{Qtz}$ scaling factors differ by only ca. 3% across the elevation transect, as compared with 18% predicted for the $^{36}Cl_{Fe}/^{10}Be_{Qtz}$ ratio. We will include some discussion of this in the revised manuscript.

In a broader sense, our view is that if we are going to try to evaluate a reaction-specific scaling model, all elements of the model should be reaction-specific for the comparison to be internally consistent.

Here I provide detailed comments, suggestions and minor corrections:

Introduction:

Paragraph starting with line 35 (or the one starting with line 48): please note that in Schimmelpfennig et al (2011) (doi:10.1016/j.quageo.2011.05.002), we have measured

36Cl/3He/21Ne on an altitude transect of Kilimanjaro. It is to my knowledge the only other altitude transect including 36Cl on a similar altitude range as your samples, and might therefore be considered to be mentioned (although 36Cl was measured in Ca-pyroxene and -plagioclase).

We have added this reference (Line 37).

Line 40: as you consider the high-energy reactions, and to avoid confusion, maybe better something like "because these reactions are sensitive to the lower end of the high-energy spectrum"?

We have adopted this wording (Line 41).

Line 105: regarding the estimated thickness of the Mt Evans sample add the reference to Table 1. Also, 25 cm is quite deep. I recommend to clarify whether or not other thicknesses would have an impact on the results. What is the reason of approximating the boulder geometry as a cube? In contrast to the other sites, the exposure history of this boulder is not mentioned here - in Table 1 and later in the text it says "steady state erosion", please give according information here in the text.

In the absence of direct constraints, a cube is a relatively straightforward and simple option. It is unlikely that deviation of the actual geometry from a perfect cube would have a significant impact on the results of the analysis. This is because we are considering only production ratios (i.e., the $^{36}$Cl production rate from Fe is cross calibrated against $^{10}$Be in quartz and $^{36}$Cl in feldspar). This implies that the sample geometry only matters to the extent that the different production mechanisms have different attenuation lengths or radioactive decay is important. We include a brief discussion of this in the text:

"The uncertainty introduced by this approach is unlikely to significantly affect the results of the analysis because production ratios are sensitive to sample thickness only insofar as different production mechanisms have different attenuation lengths or radioactive decay is significant." (Lines 105-107)

We also add to this section that we model the Mt. Evans sample as eroding in steady state (Line 103).

Lines 161-169: is there a specific reason why bulk rock is treated differently for the major element analysis?

We used a fusion approach for bulk rock so that we could determine the Si concentration, which forms a volatile fluoride. We now explain this in the text:

"However, this approach cannot be used to determine Si, which forms a volatile fluoride. To determine Si and other major element concentrations..." (Line 153-156)

If you have the measurements of all major elements in the target fraction, I recommend to add them in your supplement table, as these data are included in and necessary for the calculation of the negative muon capture yield in other calculators (Greg's calculator, GREp 36Cl, and an updated version of my 36Cl spreadsheet published in 2009).

Unfortunately, we do not have analyses of all major elements in the target fraction. However, we believe that this is likely not a major source of error in the analysis.

This is because negative muon capture is likely a minor source of $^{36}$Cl in the feldspar, accounting for ca. 1.5% of total production in the Owens Valley feldspar. There is likely no production of $^{36}$Cl by slow-negative muon capture on Fe because the average ca. 20 MeV nuclear excitation of a muon capture is insufficient for the reaction.

Lines 188-192: I recommend to add here a reference to Supp Table "Computed Parameters" (Sub-table "Fraction of 36Cl production in feldspar by target element").

We have added:

(Supp. Table "Computed Parameters") at Line 277

Line 199: I guess this should be changed to "In the constant exposure model"

We have changed this at line 174.

Line 200: for completeness, I guess you should add "...feldspar, multiplied by the total 36Cl production rate in feldspar, minus production..."

We have added this at lines 175-176.

From section 3.2 on, it would be very helpful for readers if the "types" of production rates or scaling factors were better specified each time, even if this means repetitions. E.g. in line 201 "...derive the Fe production rate at the sample site." – If I understand correctly, in line 203, P_36,i are the local production rates, right?

We have specified that the production rate here is the production rate at the site (Line 177, 180)

Line 203: as these concentrations are corrected for the radiogenic component, I would not call them "measured" here. Maybe "cosmogenic".

We have changed this to "cosmogenic" (Line 179)

Line 225: this correction is not included in Eq. 3. Shouldn't it?

It is somewhat difficult to express the steady-state erosion calibration in a succinct form. Eqn. 3 is a general equation that is meant to show how we determine erosion rates. It is applicable to both K-feldspar and magnetite.

The summation is over all relevant production mechanisms, so solving this equation for $^{36}$Cl production from Fe, as we say we do in the text, would entail subtracting the other production mechanisms.

We prefer to keep the current formation because it is concise (i.e., we do not need to have one equation for the forward calculation of the erosion rate with K-feldspar, and another for the production rate from magnetite). The main point we want to convey here is that the attenuation length matters, and, because the attenuation lengths for muon production are long (and therefore E/lambda is small), the radioactive decay can also matter.

To make it clear that the various minor production mechanisms are considered we add at lines 186-189:

"The erosion rate is then used in conjuncture with $N_{36,mt}$, and production rates from pathways other than spallation on Fe to again solve Eqn. 3, but this time for the site-specific production rate of $^{36}$Cl from Fe."

Line 258 "…in the subsurface, i.e. with increasing pressure, than 10Be in quartz (as modeled in Figure 2)…"?

We have changed this to:

"Production of $^{36}$Cl from Fe thus declines more rapidly with increasing mass-depth in the subsurface than $^{10}$Be in quartz and much more rapidly than $^{36}$Cl from K." (Line 226-227)

Line 265 "…36Cl by low-energy neutron capture by Cl…"

We have changed this to:

"by low-energy neutron capture by $^{35}$Cl" (Line 231)

Lines 271-272: please specify if following this analysis, was a correction done for low-energy neutron fluxes on the Owens Valley magnetite samples.

Yes, for the subsequent calculations we used the adjusted low-energy neutron fluxes. Added (Line 239):

"We use these adjusted values for our production rate and ratio calibrations for the Owen's Valley site"

Lines 275-286: Things become quite complicated here. To allow readers to better follow the approach, I recommend to be more specific about which rates or scaling factors come from your measured data, from modelling or from published LSD-scaled SLHL production rates. E.g. line 276: "…normalized by the LSD-scaled SLHL production rates (Borchers et al., 2016) of either 10Be in quartz…" (if that's correct).

We have tried to clean up the presentation of the normalization scheme and make more explicit where the normalization factors come from. The revised text reads (lines 242-259):

"To examine changes in $^{36}Cl_{Fe}/^{10}Be_{qtz}$ and $^{36}Cl_{Fe}/^{36}Cl_K$ with altitude, the calibrated production rates at the study sites are first normalized by the production rate of either $^{10}Be$ in quartz or $^{36}Cl$ from K at the samples' locations, whichever the calibration was conducted against, giving production ratios. To normalize differences in geomagnetic scaling between the sites, the production ratios at the calibration sites are then divided by the production ratios at sea level at their geographic locations. In this normalization scheme, all scaling factors are calculated using the scaling model of Lifton et al. (2014). Reference SLHL production rates are taken from Marrero et al. (2016a) for $^{10}Be$ in quartz and $^{36}Cl$ from K. The SLHL production rate of Fe is taken from the inverse-error weighted average SLHL production rate of the calibration samples (Table 2), excluding OV19-1 as an outlier (section 3.2). The total normalization can be expressed as:

$$R_{Fe/K,Be} = \left(\frac{P_{Fe,cal}}{S_{K,Be}*P_{K,Be,SLHL}}\right) / \left(\frac{S_{Fe,0}*P_{Fe,SLHL}}{S_{K,Be,0}*P_{K,Be,SLHL}}\right) = \left(\frac{P_{Fe,cal}}{S_{K,Be}}\right) / \left(\frac{S_{Fe,0}*P_{Fe,SLHL}}{S_{K,Be,0}}\right)$$    Eqn. 4

where $R_{Fe/K,Be}$ is the resulting altitude scaling factor ratio, $P_{Fe,cal}$ is the production rate of $^{36}Cl$ from Fe at the calibration site (as calculated from the data using the approach outlined in section 2.4.1), $S_{K,Be}$ is the scaling factor for production of $^{36}Cl$ from K or $^{10}Be$ in quartz from SLHL to the site, $S_{K,Be,0}$ is the scaling factor for production of $^{36}Cl$ from K or $^{10}Be$ in quartz from SLHL to sea level at the site, and $S_{Fe,0}$ is the scaling factor for $^{36}Cl$ production from Fe from SLHL to sea level at the site. All scaling factors are reaction-specific and are calculated following Lifton et al. (2014). Finally, $P_{Fe,SLHL}$ is the production rate of $^{36}Cl$ from Fe at SLHL, which is determined from the average of all calibration data. The SLHL production rate of $^{36}Cl_K$ or $^{10}Be_{qtz}$ ($P_{K,Be,SLHL}$) is in both the numerator and denominator and thus cancels. This

approach allows us to examine whether deviations in production ratios from the average ratio are consistent with the modeled scaling behavior by comparing calibrated scaling factor ratios with model predictions using statistical hypothesis testing and goodness-of-fit metrics."

Line 180: "...PFe,cal is the calibrated production rate of 36Cl from Fe at the calibration site (as calculated in section 3.2)".

Changed to (Line 251-252):

"$P_{Fe,cal}$ is the production rate of $^{36}Cl$ from Fe at the calibration site (as calculated from the data using the approach outlined in section 2.4.1)"

Are S_K,Be, S_K,Be,0 and most importantly S_Fe,0 LSD scaling factors or where do they come from?

We now explain that they come from the Lifton scaling model (Line 246) -

"all scaling factors are calculated using the reaction-specific scaling model of Lifton et al. (2014)"

Also, it would be helpful to add in Eq. 4 the calculation where the SLHL production rates of 36ClK or 10Beqtz are shown in both the numerator and denominator (in addition to the calculation where both are canceled).

We have added this to Eqn. 4 (Line 250)

BTW, in supp table "Computed Parameters", Marrero et al 2016" are cited for the SLHL production rates. The rates are very similar, but the citation should be consistent.

We have changed this reference to Marrero et al., 2016 (Line 247).

Line 288: correct to 36Cl_Fe/10Be_Qtz

We have corrected this (Line 284).

Fig. 1:

- In my opinion, figure captions should have a general title and not start with the description of individual panels. Here maybe something like "Comparison of high-energy particle flux energy spectra (grey curves, left y-axis) with excitation functions (dashed curves, right y-axis). A. ..."

We have added this general title to the figure caption.

- Reading of the figure would be easier/faster if you could somehow visually relate the energy spectra with the left y-axis and the excitation functions with the right y-axis, or at least give the information early in the caption (maybe similar to my suggestion above).
- In the legend: either add or remove the masses of the target elements for all reactions

We have removed the target masses.

- Millibarns is a unit, therefore better label the right y-axis "reaction cross section (millibarns)"

We have changed this label to "Reaction cross section (millibarns)"

Table 1: Please add the names of the locations (Owens Valley etc) above each batch of samples

We have added the location names to the table above each group of samples.

Fig. 3: The caption says that all sites have similar exposure ages (no erosion etc.), however, in Table 1 steady-state erosion is given for the exposure model of the Mt Evans sample. Please correct this in the caption (or in Table 1?).

We have removed this from the caption

Fig. 5:

- Please add the site names on the figure.

We have added the site names.

- Legend: next to the blue mean correct "Site mean, Fe/Qtz", or even better be consistent with the labels next to the blue and orange lines in the legend.

We have corrected this and changed the legend labels to a uniform style (starting with the ratio and then explaining what is shown).

- The same should be corrected for the grey circle and diamond in the legend

Table 2:

- Make sure that the table is easier to read in the printed paper, i.e. two columns belong to one header

We have separated the header row from the labels with a solid line. We will try to ensure that the type set table looks good and is reasonable easy to read.

- Specify somewhere (in the caption or in the headers of the columns) that these are 36Cl production rates from Fe

We have added this information to the caption.

- Here and in all supp tables, I would clarify that you exclude OV19-1 from your results

We have added a footnote here, some text in the caption to Fig. 5, and a note in supplementary Table 5 "Computed Parameters" that this sample is excluded as an outlier.

Supp Table "AMS Data":

- In the columns with the rare/stable isotope ratios, the order of magnitude is missing (I guess it must be x10^-15)

We have added x10$^{-15}$.

- For 36Cl Blank-3, two measured ratios are given for 36Cl/Cl and 35/37, respectively. So, are there actually 2 blanks? Only one mass of spike Cl mass is listed, though? Is the other mass missing? If there are different blanks (one with the felspars and one with the magnetite), wouldn't it be easier to call them the differently? Why give them the same name?
- In general, I find this table hard to read. It would be easier if you'd just list the different minerals for each sample one below the other and make one column per spike mass and measured ratio.

We have added the missing blank and reorganized the table to include the different minerals listed one below the other.

Supp Tables "Nuclide Concentrations", "Target Chemistry" and "Rock Chemistry":

- Would it be possible to add the concentrations for BL and LL here, for completeness, even if they have been published earlier?

We have added these concentrations to the supplementary tables, below the new data from this study.

Supp Table "Computed parameters":

- In the uppermost sub-table (and in the sub-table "Production Rates"), please be consistent in the headers of your columns: in the scaling factor columns you mix target elements (for 36Cl) and the produced nuclide (Be). Ideally put both, e.g. 36Cl_Fe etc and 10Be_qtz.

We have changed this to use consistent symbology (i.e., $^{36}Cl_{Fe}$, $^{36}Cl_{Ca}$, $^{10}Be_{qtz}$ etc.)

- Please explain what "Scaling factors to site at sea level" is. I'm confused to what this corresponds in your calculations and in the manuscript. X

The "scaling factors to site at sea level" are the $S_0$ values used to normalize for geomagnetic variability between sites. We have added an explanatory note in the supporting information that this refers to $S_0$ in Eqn. 4. We have also tried to improve the explanation in the main text (Lines 244-245):

"To normalize differences in geomagnetic scaling between the sites, the production ratios at the calibration sites are then divided by the production ratios at sea level at their geographic locations."

- It's unclear to me why you differentiate superscripts 2 and 3 (i.e. "production rate of 36Cl in magnetite by pathways other than spallation on Fe (Owens Valley only, used for Fe calibration)" and "production rate of 36Cl in magnetite by spallation on K, Ca, and Ti (Mt. Evans only).")

We differentiate between these two as input data for the calibration codes. For the constant-exposure samples we simply subtract the production rates of the various minor production pathways (averaged over the sample thickness and appropriately scaled) from the calibrated production rate in magnetite to get the Fe production rate.

For the Mt. Evans sample the situation is more complicated, because the different attenuation lengths of the different production mechanisms are also important. For this sample, we treat the low-energy neutron capture separately to capture this effect.

- I would simplify the values in column M (radiogenic 36Cl…) to "atoms 36Cl g-1", as that's the number that you probably use to correct your measured 36Cl concentration.

We prefer to use atoms $^{36}$Cl µg Cl$^{-1}$ here because it is the same number for both feldspar and magnetite from the same rock, whereas $^{36}$Cl g$^{-1}$ differs between feldspar and magnetite because they have different Cl concentrations.

- I recommend adding the nuclide after "atoms" in columns O, Q etc (in all sub-tables); i.e. atoms 36Cl g-1 yr-1 etc.

We have added atoms in columns O, Q, etc.

- Sub-table "Fraction of 36Cl production in feldspar by target element": Please clarify whether or not the fraction of 36Cl produced from Cl includes the radiogenic component. If yes, I get similar results for sample OV19-13 (the only one I tested that far); if not, the spallation component becomes more dominant by 15% according to my calculations. X

This calculation does not include the radiogenic component but does include the higher thermal neutron production estimated due to snow cover, which could account for a discrepancy of ca. 15% (section 2.4.4).

We have added "note 12" to supporting Table 5 "Computed Parameters" explaining this:

"Fraction of production from Cl does not include radiogenic neutron capture but does include an adjustment to the thermal neutron flux to account for snow (section 2.4.3)."